# Overall Observed Survival of Female Breast, Cervical, Colorectal, and Prostate Cancers in Antigua and Barbuda, 2017–2021: Retrospective Data from Four Study Sites

**DOI:** 10.3390/ijerph22020235

**Published:** 2025-02-07

**Authors:** Andre A. N. Bovell, Adrian Rhudd, Jabulani Ncayiyana, Themba G. Ginindza

**Affiliations:** 1Discipline of Public Health Medicine, School of Nursing and Public Health, University of KwaZulu-Natal, Durban 4000, South Africa; ncayiyanaj@ukzn.ac.za (J.N.); ginindza@ukzn.ac.za (T.G.G.); 2Urology Department, Sir Lester Bird Medical Centre, Saint John’s 268, Antigua and Barbuda; adrian.rhudd@msjmc.org; 3Cancer & Infectious Diseases Epidemiology Research Unit (CIDERU), College of Health Sciences, University of KwaZulu-Natal, Durban 4000, South Africa

**Keywords:** Antigua and Barbuda, survival analysis, Kaplan–Meier analysis, breast cancer, cervical cancer, colorectal cancer, prostate cancer

## Abstract

Understanding cancer survival is important for countries such as Antigua and Barbuda, where female breast, cervical, colorectal, and prostate cancers are burdensome to the healthcare system. This study therefore aimed to estimate the survival probabilities of patients diagnosed with these cancers between 2017 and 2021. A retrospective analytical study design was used to evaluate cancer cases abstracted from medical records at key study sites. Estimates of observed survival probabilities were determined using a Kaplan–Meier analysis. Significant differences between survival curves were assessed using the log-rank test. Hazard ratios were calculated using Cox regression. A *p*-value < 0.05 indicated significance. A total of 391 diagnosed cases were included in this study (2017–2021): female breast cancer accounted for 42%, cervical cancer accounted for 10%, colorectal cancer accounted for 20%, and prostate cancer accounted for 28%. Overall, the mean age of the participants was 61.5 (±12.9) years; 62% were female, 73% were aged > 55 years, 56% were from St. John’s, and 82% were alive at the end of 2021. The median overall survival (years) was 4.8 for female breast cancer, 4.1 for cervical cancer, 4.5 for colorectal cancer, and not reached for prostate cancer. The cancer-specific overall observed 5-year survival probabilities were 44.9% for female breast cancer, 10.8% for cervical cancer, 19.6% for colorectal cancer, and 69.0% for prostate cancer. Significant associations between disease stage and overall survival were observed in female breast and colorectal cancers. This study provides important evidence for the 5-year observed survival probabilities of the studied cancers. Healthcare improvements that support cancer survival are required.

## 1. Introduction

Cancer burden represents a serious public health challenge in most countries worldwide because of population growth and aging [1,2,3]. The occurrence of cancer is a hindrance to rising rates of life expectancy in a number of countries [1,2]. Cancer is among the first to fourth common causes of death for persons under 70 years [1,2]. Globally, there were an estimated 20 million new cancer cases and about 10 million cancer deaths in 2022 [1]; these figures remain almost unchanged from the 2020 estimates [2] but demonstrate a notable increase from the 2018 estimates of 18.1 million new cases and 9.6 million cancer-related deaths [3]. In 2022, the combined age-standardized incidence rates for all cancers worldwide were 212.5 and 186.2 for men and women, respectively, while the combined age-standardized mortality rates for men and women were 109.7 and 76.8 per 100,000, respectively [1]. Aside from highlighting the burden that cancer imposes on countries, recent global estimates of incidence and mortality are suggestive of fluctuations in incidence and deaths when compared to 2020 and 2018 estimates [1,2,3].

Concerning Latin American and Caribbean countries, cancer was responsible for 47% of the 1.2 million deaths in 2008, with a reported 4 million new cases and 1.4 million deaths in 2020 [4].

The reports of GLOBOCAN cancer incidence and mortality reports of 2018, 2020, and 2022 have consistently suggested that breast cancer in women, colorectal, cervical, and prostate cancers are ranked in the top ten cancers with respect to new cases and cancer deaths in countries worldwide [1,2,3]. Globally, these four cancers collectively were responsible for roughly 32% of cancer incidence and 23% of cancer deaths in 2020 [2], and 32% of cancer incidence and 24% of cancer deaths in 2022 [1]. Despite their increasing incidence, as is the case in regions such as the Caribbean [1], cancer survival outcomes have also improved in many countries worldwide [5], with several studies proposing reasons for survival improvements, including nationwide screening programs and the increased use of advances in drug treatment [5], especially because the early diagnosis of cancer generally increases the chances of both successful clinical management of the disease and survivorship [6]. Moreover, these reasons, together with a detailed assessment of prognostic factors, such as patient age and general state of health at presentation, health-seeking behavior, tumor type and size, tumor extent, disease stage, lymph node status, evidence of metastases, evidence of comorbidity, postoperative complications, and the availability of important therapeutic options, generally lend to an enhanced quality of care for cancer patients overall, in addition to predicting the likelihood of disease recurrence and/or death [7,8,9,10].

In Antigua and Barbuda, an English-speaking Caribbean state, between 2001 and 2005, 15% of deaths were due to cancer, in comparison to 11% from 1984 to 1989 [11,12,13]. Of 492 confirmed new cancer cases from 2001 to 2005, prostate, female breast, cervical, and colorectal cancers had some of the highest incidence rates [12].

Concerning cancer care, the Sir Lester Bird Medical Centre (SLBMC) is Antigua and Barbuda’s only tertiary care hospital, where most of the island’s cancer cases are diagnosed, treated, and managed. Other areas where cancer care is offered include The Cancer Centre Eastern Caribbean (TCCEC), a public–private cancer treatment center that provided radiotherapy and other services up until April 2024, and the Medical Benefits Scheme (MBS), a statutory health organization responsible for pharmaceutical care, in addition to financing healthcare through subventions and reimbursements for medical services, among other things [14,15]. Combined, these centers account for a high proportion of cancer-related care and contribute the most to the documented evidence of cancer cases in the country [16].

Despite advances in cancer care globally, the four named cancers remain an ongoing and major threat to the health system of Antigua and Barbuda, where financial resources are scarce and timely and appropriate treatments are key to improving patient survival [1,2,17]. To our knowledge, there is no published study on the measure of survival of these or other cancers in Antigua and Barbuda [12,18,19]. Thus, it is crucial to improve our understanding of the epidemiological burden of the four named cancers by addressing this absence of reliable cancer survival statistics.

This study therefore aimed to estimate the overall observed survival probabilities of patients diagnosed with female breast, cervical, colorectal, and prostate cancers in the population of Antigua and Barbuda from 2017 to 2021.

## 2. Materials and Methods

### 2.1. Study Design, Setting and Population

We conducted a retrospective analytical study that utilized some data and results previously reported in the publication “*Incidence, trends and patterns of female breast, cervical, colorectal and prostate cancers in Antigua and Barbuda, 2017–2021: a retrospective study*” (Bovell et al., 2025) [20], to now examine the matter of observed survival for these four cancers affecting Antigua and Barbuda.

We therefore considered cases of patients > 18 years old with a primary diagnosis of female breast, cervical, colorectum, and prostate cancers between 1 January 2017 and 31 December 2021 [20]. No patients with recurrent disease were considered for this study [20].

### 2.2. Data Collection and Management

Data collection, as described elsewhere [20], focused on diagnosed cases identified from medical records at the departments of Pathology, Urology, and Oncology, SLBMC, for persons with each cancer type categorized using the International Classification of Diseases Tenth Edition (ICD-10) codes [20,21] (C50 for breast, C53 for cervical cancer, C18, C19, and C20 for colorectal cancer, and C61 for prostate cancer) [20,21,22].

Similar cases of these patients managed at TCCEC and MBS were also identified. Consecutive cases logged between January 2017 and December 2021 were identified, their medical records were examined, and demographic, clinical, pathological, and socioeconomic data were abstracted [20]. In strict compliance with data storage and privacy protocols, additional data on those who died during the 2017–2021 period were collected from the Health Information Division, Ministry of Health, Antigua and Barbuda (HID) [20].

Because of the absence of a nationwide identification system, data abstracted from study sites were cross-referenced to allow for completeness and the removal of duplicate records where appropriate, by using a mix of their Medical Benefits Scheme identification number (MBS number) for records seen at the Medical Benefits Scheme, and their hospital-generated medical patient identifier (MPI) and MBS number for cases at the SLBMC. For cases at TCCEC, we relied on both their TCCEC unique number and MBS number, and for the data on deaths, their MBS number and/or MPI.

Using a conceptual hierarchical framework [23], aggregate and cancer-specific data were classed as baseline characteristics, that is, demographic characteristics, including age at diagnosis (a continuous variable), age group (<55 years, ≥55 years old), sex (males, females), parish or district (other parishes (Barbuda, St. George, St. Peter, St. Mary, St. Phillip, and St. Paul) and St. John’s Parish), years of presentation (2017 to 2021), and vital status (alive and dead); clinicopathological characteristics, including estimates of disease stage, based on an approach discussed by Hennis et al., 2009 [24], and presented as early-stage disease (clinical stages 1 and 2 at diagnosis) and late-stage disease (clinical stages 3 and 4 at diagnosis), and had non-communicable disease other than cancer (yes, no); and socioeconomic characteristics, which included status of employment at presentation (in employment, no employment) [20].

### 2.3. Outcomes Determined

The primary outcomes determined were (i) the five-year overall observed survival probabilities for each cancer type; (ii) the observed survival probabilities for each cancer by age group, parish (district), and disease stage [25]; and (iii) the risk of overall survival for each cancer by age group, parish (district), and disease stage [6,25].

Observed survival for each case was calculated from the date of diagnosis to either the date of death or the last date of the study period, which was 31 December 2021 [6].

Because of the data available, this study assumed the following: (i) except for death, no other form of censoring occurred, and (ii) each cancer case had access to some form of standard of care or treatment available in Antigua and Barbuda in the year of diagnosis.

The overall 5-year crude point prevalence rates for the four cancers combined, along with that for each cancer type, were also estimated so as to provide added details on the number of patients who were alive at the end of the study period.

### 2.4. Data Analysis

We assessed data distribution using descriptive and summary statistics (counts, frequencies, percentages, mean and standard deviation) [6,20]. Overall and cancer-specific crude point prevalence rates were determined based on an approach discussed by Mudiyanselage et al., 2024 [26], and as per the equation$$\mathrm{Cp}=\frac{\left(I_{C}-D_{d}\right)}{\left(E_{p}\right)}$$

Ic is the total cases of all four cancer types diagnosed between 2017 and 2021;Dd is the total of all deaths of persons diagnosed with the four cancers between 2017 and 2021;Ep is the 5-year total of the projected local population at risk;Cp is the overall crude prevalence rate.

The exact Poisson confidence intervals for the crude point prevalence rates were also presented [24].

Note that the description of each variable in the equation was changed to accommodate values specific to each cancer type when needed.

Overall cumulative observed survival probabilities were determined using Kaplan–Meier analysis [6,27]. The log-rank test was used to assess differences in survival between the cumulative probabilities of the noted groups for each characteristic mentioned in the ascertainment of outcomes [6]. The risk for survival was determined using Cox proportional hazards regression analysis [27]. Statistical significance was defined as *p*-value < 0.05 in a univariate model [27].

Further, using the continuous variable age, which was common to each cancer-specific dataset and enabled the ease of examination, the distributions of all cancer-specific data were assessed visually for normality using a histogram; normality was also examined with the use of the Shapiro–Wilk test [28]. The appearance of a roughly symmetrical or bell-shaped distribution was taken as good judgment that cancer-specific data were normally distributed [28]. Additionally, a Shapiro–Wilk W test *p*-value > 0.05 provided confirmation that all cancer-specific data were normally distributed [28].

The analyses were performed using STATA software version 17/SE-Standard Edition and Microsoft Excel [20].

### 2.5. Ethical Considerations

This study has received approval from the Antigua and Barbuda Institutional Review Board, Ministry of Health (AL-04/052022-ANUIRB), the Institutional Review Board of the Sir Lester Bird Medical Centre, and the University of KwaZulu-Natal Biomedical Research Ethics Committee (BREC/00004531/2022) [20].

We did not have any direct contact with patients, and neither was there a risk posed to persons [29]. The details of all patients were kept de-identified and anonymized by not taking a record of their names [29].

## 3. Results

### 3.1. General Descriptions

Table 1 presents the overall and cancer-specific baseline details of the study population and on the assumption that the data on each cancer type were normally distributed (Appendix A). In total, 391 cases of the four cancers were diagnosed in the period 2017–2021 and were included in this study (Table 1). This approximated to an overall combined 5-year crude point prevalence rate of roughly 66.8 (95% CI: 59.7–74.6) per 100,000 population after taking into account local projected population at risk estimates (average of 95,738 per year) for the study period [30].

For the common characteristics looked at, the overall median age at diagnosis/presentation was 62 years (Table 1) [20]. Of the defined age categories, the <55 years age group accounted for 27.4% of cases, with the ≥55 years age group accounting for 72.6% of cases (Table 1). By percentage of cases, female breast cancer was responsible for 42%, prostate cancer 28%, cervical cancer 10%, and colorectal cancer 20% (Table 1) [20]. Regarding year of presentation, aside from female breast cancer in 2018, the year 2020 had the highest count of diagnosed cancer-specific cases (Table 1) [20]. With respect to parishes, St. John’s Parish was responsible for 56% of our case count (Table 1) [20].

### 3.2. Female Breast Cancer

For female breast cancer, the 5-year crude point prevalence rate was 56.1 (95%CI 47.6–66.7) per 100,000 females after taking into account local projected population at risk estimates (average of 49,900 per year) for the study period [30]. The median age at diagnosis/presentation was 59 years (Table 1) [20]. In general, 64% of cases were aged ≥ 55 years, and 36% were aged < 55 years. St. John’s Parish accounted for 48% of all cases, while the other parishes combined accounted for 52% (Table 1). Regarding disease stage, 53% of cases had early-stage disease, while the remaining 47% had late-stage disease (Table 1). There were 22 deaths, with a median survival time of roughly 4.6 years and a 5-year overall survival probability of 44.9% (standard error 12.9%) (Table 2) (Figure 1a). The mean follow-up time was 2.41 (95% CI 2.20–2.63) years. Assessing the significant differences between the survival curves (i) by age group showed that the 5-year overall survival probability in patients aged < 55 years and ≥55 years was 35.8% and 52.9%, respectively (*p*-value = 0.236) (Figure 2a); regarding parish, the 5-year overall survival in the other parishes combined and St. John’s Parish was 43.3% and 56.1%, respectively (*p*-value = 0.556) (Figure 3a); and, regarding early- and late-stage disease, the 5-year overall survival was 68.3% and 31.0%, respectively (*p*-value = 0.031) (Table 2) (Figure 4a). Except for disease stage, where the risk of death for a patient with late-stage disease was roughly 2.9 times greater than that for a patient with early-stage disease (*p*-value < 0.05), the univariate analysis of age group and parish did not show a significant association with overall survival (Table 3).

### 3.3. Cervical Cancer

For cervical cancer, the 5-year crude point prevalence rate was 10.4 (95%CI 6.8–15.3) per 100,000 females after taking into account local projected population at risk estimates (average of 49,900 per year) for the study period [30]. Our median age at diagnosis/presentation was 52 years (Table 1) [20]. Generally, 63% of cases were aged < 55 years, and 37% were aged ≥ 55 years. St. John’s Parish accounted for 68% of all cases, while the other parishes combined accounted for 32% (Table 1). Regarding disease stage, 55% of cases had late-stage disease, while the remaining 45% had early-stage disease (Table 1). There were 14 deaths, with an estimated median survival time of 4.10 years (95% CI 3.41–4.52) and a 5-year overall survival probability of 10.8% (standard error 10%) (Table 2) (Figure 1b). The mean follow-up time was 2.22 (95% CI 1.77–2.66) years. Assessing the significant differences between the survival curves (i) by age group showed that the 5-year overall survival in those aged < 55 years and ≥55 years was 30.1% and 21.4%, respectively (*p*-value = 0.915) (Figure 2b); (ii) regarding parish, the 5-year overall survival in the other parishes combined and St. John’s Parish was 39.7% and 28.5%, respectively (*p*-value = 0.713) (Figure 3b); and (iii) regarding early- and late-stage disease, the 5-year overall survival was 42.3% and 23.0%, respectively (*p*-value = 0.223) (Table 2) (Figure 4b). The univariate analysis did not show any evidence of a significant association between overall survival and age group, parish, or disease stage (Table 3).

### 3.4. Colorectal Cancer

For the 79 colorectal cancer cases, the 5-year crude point prevalence rate was 11.9 (95% CI 9.0–15.4) per 100,000 population after taking into account local projected population at risk estimates (average of 95,738 per year) for the study period [30]. Our median age at diagnosis/presentation was 67 years (Table 1) [20]. Regarding age category, 75% of cases were aged ≥ 55 years, while 25% were aged < 55 years (Table 1). Regarding parish, St. John’s Parish accounted for 56% of cases, while the other parishes accounted for 44% of cases (Table 1). The highest number of cases seeking treatment was observed in 2020 (34%), while the lowest number was observed in 2019 (9%). Regarding disease stage, 58% had late-stage disease, while 42% had early-stage disease. There were 22 deaths during the study period. The estimated median survival time was 4.52 years (95% CI 3.94–4.89), with the 5-year overall survival being 19.6% (standard error 11.3%) (Table 2) (Figure 1c). The mean follow-up time was 2.25 (95% CI 1.91–2.59) years. Estimating the significant differences between the survival curves by (i) age group showed that the 5-year overall survival in those aged < 55 years and ≥55 years was 26.5% and 21.7%, respectively (*p*-value = 0.879) (Figure 2c); (ii) regarding parish, the 5-year overall survival in the other parishes combined and St. John’s Parish was 63.8% and 15.0%, respectively (*p*-value = 0.270) (Figure 3c); and (iii) regarding disease stage, the 5-year overall survival for early- and late-stage disease was 38.4% and 24.3%, respectively (*p*-value = 0.042) (Table 2) (Figure 4c). Evidence from the univariate analysis showed a significant association between disease stage and overall survival, with the risk of dying for a patient with late-stage colorectal cancer estimated at 1.6 times higher than that for a patient with early-stage disease (*p* < 0.05) (Table 3).

### 3.5. Prostate Cancer

For prostate cancer, the 5-year crude point prevalence rate was 41.9 (95%CI 33.9–51.2) per 100,000 males after taking into account local projected population at risk estimates (average of 45,839 per year) for the study period [30]. Our median age at diagnosis/presentation was 67 years (Table 1) [20]. About 97% of cases were 55 years old, while only 3% were under 55 years old. Regarding parish, St. John’s accounted for 62%, with the other parishes combined accounting for the remaining 38% of cases (Table 1). Regarding the year of diagnosis/presentation, 37% of all cases were recorded in 2020, followed by 25% in 2019 (Table 1). Regarding disease stage, 46% had late-stage disease, while 54% had early-stage disease. There were 13 deaths. The median survival time was not reached within the defined study period. The estimated five-year overall survival was 69.0% (standard error 10.2%) (Table 2) (Figure 1d). The mean follow-up time was 2.14 (95% CI 1.90–2.39) years. Examining the significant differences between the survival curves by (i) age group indicated that the 5-year overall survival in those aged < 55 years and ≥55 years was 100% and 69.3%, respectively (*p*-value = 0.768) (Figure 2d); (ii) regarding parish, the 5-year overall survival in the other parishes combined and St. John’s Parish was 79.6% and 63.8%, respectively (*p*-value = 0.604) (Figure 3d); and (iii) regarding disease stage, the 5-year overall survival for patients with early- and late-stage disease was 76.7% and 59.3%, respectively (*p*-value = 0.313) (Table 2) (Figure 4d). While the univariate analysis of age group was undefined, for parish and disease stage, no evidence of a significant association with overall survival was found (Table 3).

## 4. Discussion

This study provides important evidence on the overall observed Kaplan–Meier 5-year survival probabilities of female breast, cervical, colorectal, and prostate cancers in Antigua and Barbuda from 2017 to 2021. The overall 5-year observed survival probabilities were as follows: 44.9% for female breast cancer, 10.8% for cervical cancer, 19.6% for colorectal cancer, and 69.0% for prostate cancer. The findings show that a female breast, cervical, or colorectal cancer case has a 50% chance of survival if they live for at least 4.75, 4.10, and 4.53 years, respectively. For prostate cancer, the median survival time was not reached within the study period. This observation could be an indication of the time to progression for men diagnosed, the probable impact of early diagnoses, or the result of a lead-time bias [31]. A future survival study covering a longer study period could potentially provide a definitive examination of lead-time and length biases in addition to a reliable estimate of the median survival time for prostate cancer.

Because of differences in survival measures, study populations, objectives, and methodologies, it is difficult to make general comparisons of the observed survival probabilities for these cancers with certain observations made in other countries [19]. Notwithstanding, the overall observed survival estimates for our cancers show that, while prostate and female breast cancer patients had a superior overall 5-year observed survival, patients diagnosed with cervical and colorectal cancers had relatively poorer overall 5-year observed survival probabilities. Although the observations for prostate cancer appear to be consistent with those in studies from North America, Europe, and the French West Indies, suggesting that there has been an overall improvement in prostate cancer survival in these regions [32,33], our observation could be an indication of the overall improvements in the management of prostate cancer in Antigua and Barbuda in recent years. These improvements include but are not limited to an increase in (i) the number of specialist physicians, urologists, and medical oncologists; (ii) the uptake of early screening and detection using prostate-specific antigen testing (PSA testing); (iii) diagnostic capabilities with access to privately performed biopsies; (iv) access to surgical and radiotherapeutic opportunities; and (v) access to systemic therapy through the Medical Benefits Scheme [18,31]. Our results, barring the country’s resource limitations, appear marginally lower than those of a study on the overall survival of prostate cancer in the French Caribbean country of Martinique, which has a well-established population-based cancer registry and a similar population composition [33].

The observed survival of female breast cancer, while consistent with generally observed trends in developing countries [19], is an indication of the level of uptake of early screening, detection, and treatment services locally [32]. While not ruling out the probable influence of length and lead-time biases [32], our findings are an indication of the level of prognosis achievable in a low-resource setting [19]. Understanding that several complementing factors could affect breast cancer survival, future studies that consider these and other determinants may provide more definitive answers [19,34]. The poor survival outcomes of cervical cancer, though consistent with observations in developing countries such as those in Latin America, the Caribbean, and Sub-Saharan Africa, appear inconsistent with general observations made in developed countries, suggesting increasing survivability resulting from the expanded role of human papilloma virus (HPV) vaccination, HPV testing, and early cervical cancer screening [35,36]. Because the prognosis of cervical cancer patients depends on the clinical extent of the disease at diagnosis, our findings suggest a possible low level of uptake of cervical cancer screening services in the country during the study period. While there is yet no published evidence on the country’s past uptake of cervical cancer screening rates with Pap smear testing, it is envisioned that, with the scaling up of local services [19], which included the introduction of HPV vaccination in 2018, HPV testing in 2022, and the establishment of the National Cervical Cancer Screening and Testing Guidelines in 2023 [37,38], Antigua and Barbuda could anticipate an improvement in survivability in the not-too-distant future. The effect of these and other programmatic interventions on the overall observed survival of cervical cancer could be borne out in future research. General observations regarding colorectal cancer, though small when compared to observations in Martinique [39,40], could be the result of a low incidence of screening-detected colorectal cancer, an indication of the late presentation of cases, limitations in deploying effective therapeutic strategies [40], and a reflection of the need for the establishment of a comprehensive national colorectal cancer screening program that enables improvements in screening and detection capabilities at the national level [38,41]. In endorsing primary prevention and the early detection of colorectal cancer, Sankaranarayanan and colleagues found that screening with a fecal occult blood test and the use of flexible sigmoidoscopy and colonoscopy led to the early detection of polyps and colorectal cancer [19], which invariably led to improved survivability over time [19].

Concerning differences between the probabilities of the survival curves for each variable of interest, our study results suggest that, for each cancer type, while patients aged 55 years or older had a lower cumulative overall observed survival, there was generally no significant difference in the cumulative overall survival between them and those aged less than 55 years, an observation that could be the effect of a probable link between aggressive disease processes and a population composed of 85% of persons of African ancestry [42,43].

Similar observations were also noted for each cancer type when examined by parish (district), a finding suggesting that this may be due to patients generally having almost equal access to cancer care across the country [44]. Aside from this suggestion, however, these observations could also be the result of chance or the natural consequences of certain inter- and intra-parish variations [45], including subtle differences in the ease of access to care and the help-seeking behavior of residents [45]. Prospective studies could be conducted to assess the effect of these and other dimensions on the cumulative observed survival outcomes of residents across geographic boundaries in Antigua and Barbuda.

Estimates of the cumulative observed survival for individuals with late-stage disease appear to be lower than that for persons with early-stage disease. Despite this observation, however, univariate Cox regression suggested that the difference in survival experience between early- and late-stage disease was only significant for patients with colorectal and female breast cancers. The findings indicate that for colorectal cancer the risk was almost 2 times greater for patients with late-stage disease, while for female breast cancer, it was 3 times greater. Generally, this observation is consistent with the findings of several studies suggesting that patients with early-stage disease who receive appropriate treatment usually have better overall survival rates than those with late-stage disease [5,46,47]. This further suggests that initiatives that encourage regular screening, early detection, and easy access to treatment could likely improve the survivability of these cancers in the future.

Because providing improvements in cancer awareness, health infrastructure, and efficiencies in health services are key to boosting the observed survival outcomes of our studied cancers, the results of this study hold promise for enhancing cancer care and treatment through justifiable health policies in Antigua and Barbuda [19,32]. This study provides the first evidence-based benchmark for locally assessing the survival of the mentioned cancers through the encouragement of both public health and clinical interventions. These interventions could include initiatives aimed at policy development, health planning, healthcare financing, effective screening and detection, and treatment [25,48].

Our study results also demonstrated that the crude point prevalence rate for female breast cancer was the highest, followed by that of prostate cancer, colorectal cancer, and cervical cancer, which might be a general reflection of the successes and/or failures of current cancer-specific diagnosis, treatment, and disease management modalities [49]. Future studies on survival may want to consider the effect of varying cancer management approaches on estimates of population prevalence. Additionally, because improved cancer survivability means increased disease prevalence, our results indicate the need for the prudent use of scarce financial resources in managing survivor care [50]. Further research on cancer survival and epidemiology in Antigua and Barbuda is also encouraged by our study results.

In general, and despite Antigua and Barbuda’s very high human development index [51,52], during the period from 2017 to 2021, the country’s public health system notably lacked an active or systematic screening program for the four cancers studied [52]. This means that any observed differences between our study estimates and those generated in the international context, for instance, as in the case of female breast and cervical cancers, could have been severely affected by issues such as (i) the absence or ineffectiveness of early screening and detection initiatives, (ii) access to timely care, and (iii) the level or quality of clinical care and management [53].

This study has several strengths. This is the first study of its kind in Antigua and Barbuda to examine the overall observed survival probabilities of four common cancers using the Kaplan–Meier method. Our study utilized cancer data from study sites named in Section 2 of this article. Collectively, these facilities routinely collect and preserve much of the documented evidence for the four cancers on the island. This highlights the importance of the data generated by these entities in deriving survival estimates at a time when a national cancer registry is absent. It also signifies the need to establish both a hospital- and population-based cancer registry on the island with the potential to enable more in-depth analyses of various dimensions of cancer survival in Antigua and Barbuda [12,18,31]. Additionally, the use of Kaplan–Meier methodology means that, despite its reliance on univariable analysis, the problem of the short study period was mitigated given its advantage of calculating survival probabilities for cancer at a given point in time within any defined period and/or context of censoring [54].

The retrospective nature of the data collection means that the study information obtained could have easily been impacted by a priori recording and/or reporting biases, especially in instances where the person(s) inputting information in patient logs or medical records did so with varying degrees of accuracy or completeness [55]. Further, our population of cases could easily have also been impacted by systematic selection bias, particularly in instances where patients who presented with an already advanced disease stage were declined diagnostic and/or therapeutic interventions, were more likely to be underreported, and may have been recommended for disease management elsewhere [56,57]. This could have had the unintended consequence of their data being excluded from the key data capture areas at our study sites, thus affecting the derivation of both overall and cancer-specific study estimates. A future study that accounts for data capture across other areas of cancer care, such as medical outpatient and hospice care, and/or that considers model adjustments to counter the possibility of selection bias may be helpful for deriving more comprehensive survival estimates [58]. Additionally, our study’s reliance on data from the previously named study sites means that it could have been affected by the smallness of the sample sizes, especially for cervical cancer cases. Any effect of these limitations on the generalizability of the study results, however, would have been attenuated because most locally diagnosed cancers are treated and/or managed by the SLBMC and the TCCEC with pharmacological and financial support from the Medical Benefits Scheme, Antigua and Barbuda. Further, the age adjustment of survival estimates was hindered by our choice of methodology, which did not include comparative survival analyses [59]. Future studies that investigate cancer survival from this perspective could be considered.

## 5. Conclusions

This study’s findings provide significant evidence on the overall 5-year observed survival probabilities of four common cancers in Antigua and Barbuda. Prostate and female breast cancers had better survival probabilities than colorectal and cervical cancers. Late-stage disease was associated with lower overall observed survival for female breast and colorectal cancer, respectively. Although we are confident that our study findings shed light on an aspect of cancer epidemiology that is lacking in Antigua and Barbuda, we believe that it would be beneficial to have an established national cancer registry, or, at the minimum, a hospital-based registry, and to consider a longer study period and more data as a means of supporting future research on cancer survival and cancer care-related treatment modalities. Additionally, our findings provide a useful benchmark for policymakers in the Ministry of Health in Antigua and Barbuda, and at the clinical practice level for implementing survival-related public health and/or clinical interventions in order to comprehensively evaluate the survival outcomes of patients screened, diagnosed, and treated for the named four common cancers in the foreseeable future.

## Figures and Tables

**Figure 1 ijerph-22-00235-f001:**
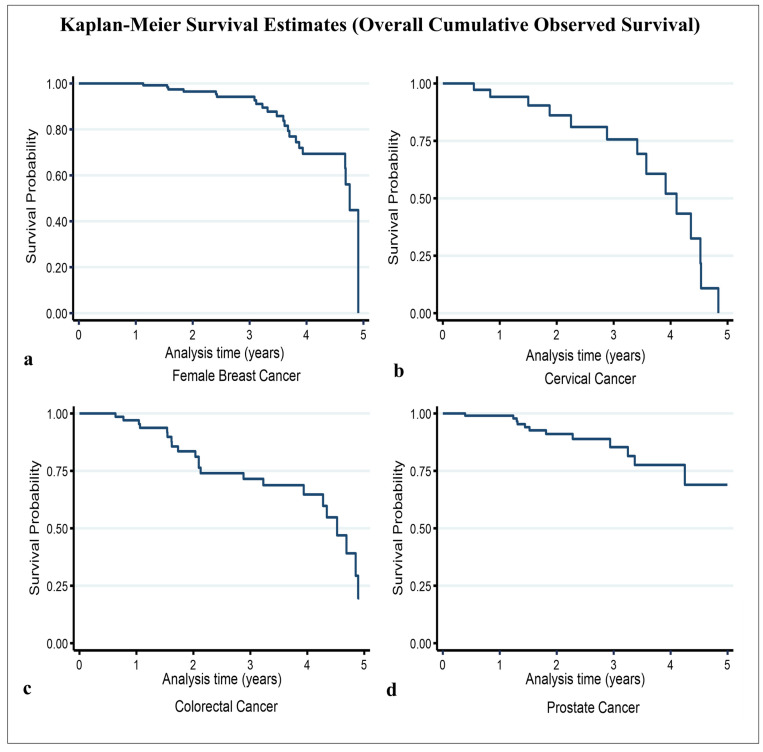
Overall cumulative observed survival by cancer type: (**a**) female breast, (**b**) cervical, (**c**) colorectal, and (**d**) prostate.

**Figure 2 ijerph-22-00235-f002:**
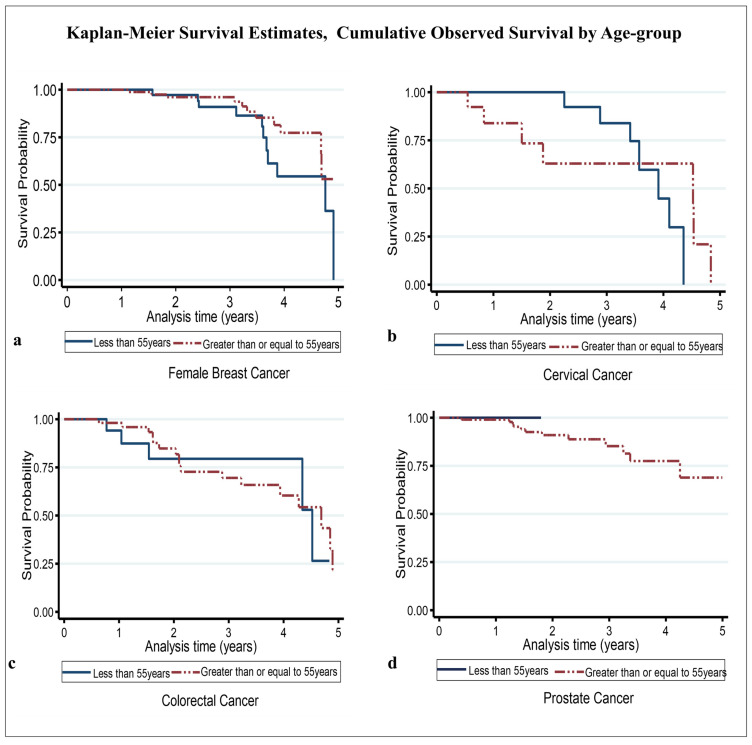
Overall cumulative observed survival by age group: (**a**) female breast, (**b**) cervical, (**c**) colorectal, and (**d**) prostate.

**Figure 3 ijerph-22-00235-f003:**
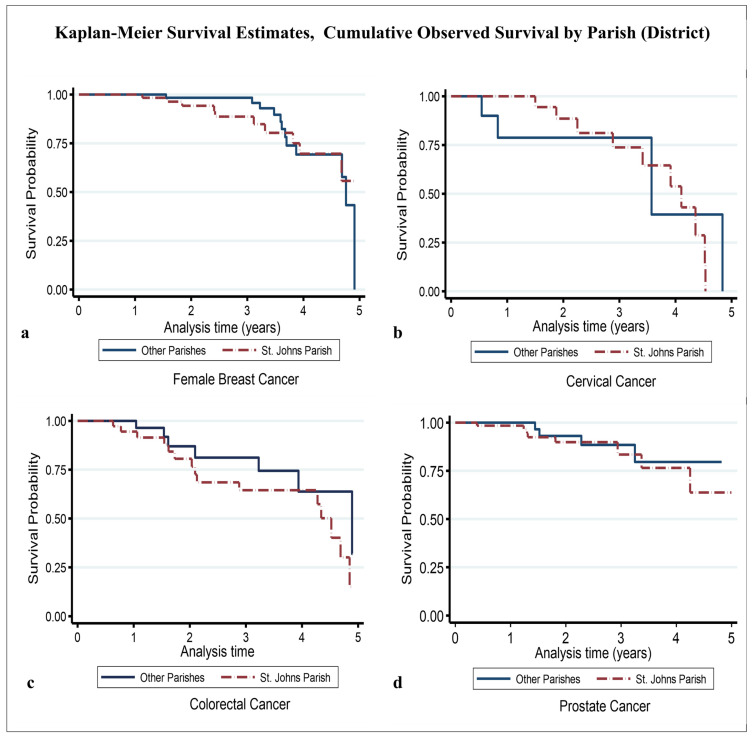
Overall cumulative observed survival by parish (district): (**a**) female breast, (**b**) cervical, (**c**) colorectal, and (**d**) prostate.

**Figure 4 ijerph-22-00235-f004:**
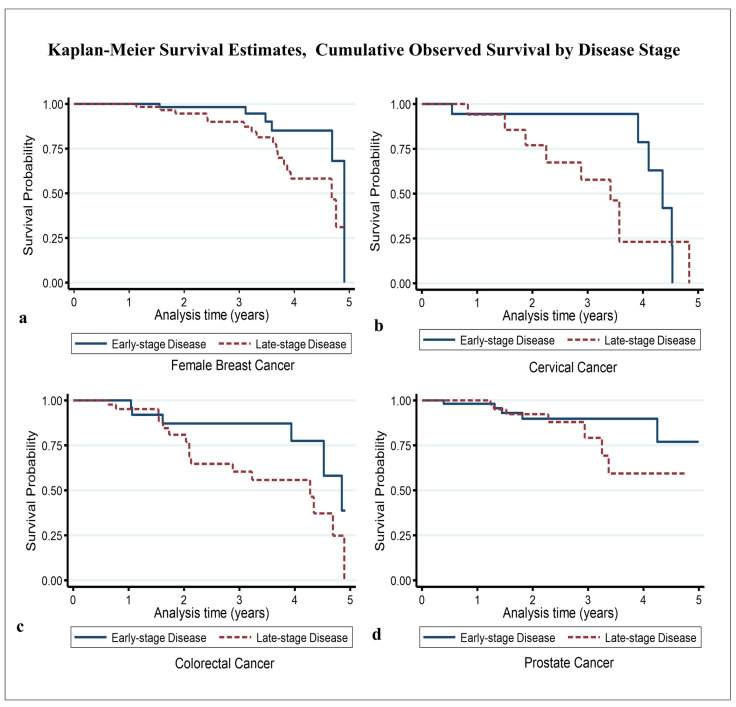
Overall cumulative observed survival by disease stage: (**a**) female breast, (**b**) cervical, (**c**) colorectal, and (**d**) prostate.

**Table 1 ijerph-22-00235-t001:** Some key attributes of our population of cases of the four cancers under study in Antigua and Barbuda: 2017–2021.

Key Attributes	The Four Cancers Combined n = 391 n (%)	Female Breast Cancer (n = 163); 42% n (%)	Cervical Cancer (n = 40); 10% n (%)	Colorectal Cancer (n = 79); 20% n (%)	Prostate Cancer (n = 109); 28% n (%)
Age at Diagnosis/Presentation					
Mean age (±SD)	61.5 (±12.9)	58.7 (±13.5)	51.8 (±15.0)	65.2 (±12.1)	66.5 (±7.8)
Mean age 95%CI	60.2–62.8	56.6–60.8	47.0–56.6	62.5–67.9	65.0–67.9
Median age (IQR)	62.0 (17.0)	59.0 (19.0)	51.5 (16.0)	67.0 (20.0)	67.0 (11.0)
Age categories					
<55 years	107 (27.4)	59 (36.2)	25 (62.5)	20 (25.3)	3 (2.7)
≥55 years	284 (72.6)	104 (63.8)	15 (37.5)	59 (74.7)	106 (97.3)
Sex					
Females	244 (62.4)	163 (100.0)	40 (100.0)	41 (51.9)	0
Males	147 (37.6)	0	0	38 (48.1)	109 (100.0)
Parish (district)					
Other parishes combined	173 (44.2)	84 (51.5)	13 (32.5)	35 (44.3)	41 (37.6)
St. John	218 (55.8)	79 (48.5)	27 (67.5)	44 (55.7)	68 (62.4)
Year of Diagnosis/Presentation					
2017	61 (15.6)	26 (16.0)	9 (22.5)	15 (19.0)	11 (10.1)
2018	66 (16.9)	36 (22.1)	6 (15.0)	13 (16.5)	11 (10.1)
2019	74 (18.9)	34 (20.9)	6 (15.0)	7 (8.9)	27 (24.8)
2020	114 (29.2)	35 (21.5)	11 (27.5)	27 (34.2)	40 (36.7)
2021	76 (19.4)	32 (19.6)	8 (20.0)	17 (21.5)	20 (18.4)
Vital Status					
Alive	320 (81.8)	141 (86.5)	26 (65.0)	57 (72.2)	96 (88.1)
Dead	71 (18.2)	22 (13.5)	14 (35.0)	22 (27.9)	13 (11.9)
Disease Stage at Diagnosis					
Early-stage	na	87 (53.4)	18 (45.0)	33 (41.8)	59 (54.1)
Late-stage	na	76 (46.6)	22 (55.0)	46 (58.2)	50 (45.9)
Had Non-communicable Disease Other than Cancer					
No	222 (56.8)	82 (50.3)	29 (72.5)	44 (55.7)	67 (61.5)
Yes	169 (43.2)	81 (49.7)	11 (27.5)	35 (44.3)	42 (38.5)
Status of Employment at Presentation					
No employment	122 (31.2)	51 (31.3)	9 (22.5)	33 (41.8)	29 (26.6)
In employment	269 (68.8)	112 (68.7)	31 (77.5)	46 (58.2)	80 (73.4)

na—not applicable.

**Table 2 ijerph-22-00235-t002:** Cancer-specific 5-year observed survival plus cumulative 1-year, 3-year, and 5-year survival probabilities by variables of interest for each cancer type.

Cancer Type	Variables of Interest	n	1-Year (%)	3-Year (%)	5-Year (%)	Log-Rank *p*-Value	Cancer-Specific 5-Year Observed Survival/SE (%)
Female Breast	Age group					0.236	**44.9 (12.9)**
	<55 years	59	99.6	90.7	35.8		
	≥55 years	104	99.6	95.8	52.9		
	Parish					0.556	
	Other parishes	84	100.0	97.9	43.3		
	St. Johns	79	97.9	88.6	56.1		
	Disease Stage					**0.031**	
	Early-stage	87	100.0	98.4	68.3		
	Late-stage	76	98.4	90.0	31.0		
Cervical	Age group					0.915	**10.8 (10.0)**
	<55 years	25	100.0	84.0	30.1		
	≥55 years	15	84.0	62.6	21.4		
	Parish					0.713	
	Other Parishes	13	78.6	78.6	39.7		
	St. Johns	27	99.6	73.4	28.7		
	Disease Stage					0.223	
	Early-stage	18	94.2	94.2	42.3		
	Late-stage	22	94.2	58.1	23.0		
Colorectal	Age group					0.879	**19.6 (11.3)**
	<55 years	20	93.9	79.5	26.5		
	≥55 years	59	98.1	69.6	21.7		
	Parish					0.270	
	Other parishes	35	100.0	80.7	63.8		
	St. Johns	44	94.1	64.5	15.0		
	Disease Stage					**0.042**	
	Early-stage	33	100.0	87.2	38.4		
	Late-stage	46	94.9	60.5	24.3		
Prostate	Age group					0.768	**69.0 (10.2)**
	<55 years	3	100.0	NA	NA		
	≥55 years	106	99.1	85.4	69.3		
	Parish					0.604	
	Other parishes	41	100.0	88.3	79.6		
	St. Johns	68	98.3	83.1	63.8		
	Disease Stage					0.313	
	Early-stage	59	98.4	90.0	76.7		
	Late-stage	50	99.8	79.5	59.3		

SE—standard error.

**Table 3 ijerph-22-00235-t003:** Results of univariate Cox regression analysis of age group, parish, and disease stage by cancer type.

Cancer Type	Variables of Interest	Hazards Ratio	95% Confidence Interval	*p*-Value
Female Breast	Age group (<55 years vs. ≥55 years)	0.596	0.251–1.414	0.240
	Parish (other parishes vs. St. Johns)	1.295	0.546–3.074	0.557
	Disease stage (early-stage vs. late-stage)	2.886	1.054–7.904	**0.039**
Cervical	Age group (<55 years vs. ≥55 years)	0.934	0.259–3.364	0.915
	Parish (other parishes vs. St. Johns)	1.279	0.344–4.746	1.279
	Disease stage (early-stage vs. late-stage)	2.011	0.642–6.298	0.230
Colorectal	Age group (<55 years vs. ≥55 years)	0.924	0.335–2.550	0.879
	Parish (other parishes vs. St. Johns)	1.656	0.669–4.100	0.276
	Disease stage (early-stage vs. late-stage)	1.614	1.002–2.600	**0.049**
Prostate	Age group (<55 years vs. ≥55 years)	-	-	-
	Parish (other parishes vs. St. Johns)	1.373	0.412–4.573	0.606
	Disease stage (early-stage vs. late-stage)	1.810	0.563–5.825	0.320

## Data Availability

All data generated or analyzed during this study are included in the article. Data are fully available without restrictions and inquiries can be directed to the corresponding author.

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
