# Peer review of "Overall Observed Survival of Female Breast, Cervical, Colorectal, and Prostate Cancers in Antigua and Barbuda, 2017–2021: Retrospective Data from Four Study Sites"

_ijerph, 2025, doi:10.3390/ijerph22020235_

Round 1

Reviewer 1 Report

Comments and Suggestions for Authors

General

The paper of Bovell et al. focused on the 5-year survival of female breast, cervical, colorectal and prostate cancer patients in Antigua and Barbuda as a descriptive study. Since cancer deaths are meaning one of the highest mortality burdens globally, the aim of the study seems actual and relevant. In addition, relatively limited data are available that depicts cancer epidemiology of the country. Nevertheless, based on the manuscript, numerous questions arose which should be answer to help general quality of the work.

Major and minor remarks

·         Page 1, line 35-45: The authors cited previously published data of GLOBOCAN from the years of 2018 and 2020. The most recent publication incorporates data of 2022; thus, CA Cancer J Clin. 2024 May-Jun;74(3):229-263 should be referred also.

·         Page 2, line 80: The current study focused on the patient material of three institutes, e.g. Sir Lester Bird Medical Centre (SLBMC), The Cancer Centre Eastern Caribbean (TCCEC), and the Medical 81 Benefits Scheme (MBS). If this is a hospital-based assessment, it should be noted in the title of the manuscript. On the other hand, Results section did not describe the coverage of the study compared to the whole domestic population. Note that according to GLOBOCAN 2022, in Barbados ~362 new cases estimated per 100,000 in a single year, while the recent study refers similar amount of cases for 5-year period in Antigua and Barbuda.

·         Thus, to avoid further misunderstanding, the Introduction section should share information about the healthcare system of Antigua and Barbuda.

·         According to page 2, line 85, the authors analyzed only the histologically verified cases. However, usually in case of patients with the most advanced stage diagnostic and therapeutic interventions are not performed. In that case the least favorable outcome cases were excluded from the analysis; therefore, the observed survival may be better than real. Firstly, it should be described as on of the limitations. Secondly, the authors should provide information about number of cases with the targeted ICD-10 codes without histological verification. In addition, rate of morphologically verified cases is an important quality assurance parameter for cancer registration.

·         Page 3, line 96: defining of the applied ICD-10 codes seemed incomplete. Not only C53.9 is related to cervical cancer, but the whole C53 group. C19 (malignant neoplasm of rectosigmoid junction) is absent from the colorectal cancer group, only C18 and C20 was mentioned.

·         Based on the Methods section, it was not clear that how persons were identified correctly, how duplications were avoided (e.g. patients were examined or treated in different departments), and how mortality database was linked to hospital data. Is there any unique person ID in the country?

·         What was the date of last follow-up when vital status was recorded?

·         The authors provided 5-year survival, however, in case of patients diagnosed in 2020 and 2021 it was not complete. Censored cases of those years may impair further survival rate in 2025 and 2026 in sum.

·         Page 6: According to Table 2, the observed 5-year overall survival of breast cancer was 44.9%, which is extremely low compared to that of countries with very high human development index. In the literature 70-90% was found, while according to SEER, survival of breast cancer patients with earlies stage exceeds 99% (https://www.cancer.org/cancer/types/breast-cancer/understanding-a-breast-cancer-diagnosis/breast-cancer-survival-rates.html). What is the rationale between these huge differences? Also, the survival of cervical and colorectal cancer showed unfavorable outcome in international context, while prostate cancer was in similar range or mildly exceeded the survival of other countries.

·         The authors should cite international examples for population-based cancer survival studies in the Discussion.

·         The recent cancer screening system in Antigua and Barbuda may explain some findings of the study.

Summary

Albeit this paper contains requires major reconsideration, the performed work may be of great interest for the professional audience.

Author Response

Response to Reviewer 1 Comments

Thank you very much for taking the time to review this manuscript. We do express our appreciations to you for your comments and suggestions offered. It is our hope that the revised manuscript has addressed your concerns. We do look forward to hearing from you on this.

Please find the detailed responses below and the corresponding revisions/corrections highlighted in track changes in the re-submitted files.

2. Point-by-point response to Comments and Suggestions for Authors

REVIEWER 1

General comments:

General

The paper of Bovell et al. focused on the 5-year survival of female breast, cervical, colorectal and prostate cancer patients in Antigua and Barbuda as a descriptive study. Since cancer deaths are meaning one of the highest mortality burdens globally, the aim of the study seems actual and relevant. In addition, relatively limited data are available that depicts cancer epidemiology of the country. Nevertheless, based on the manuscript, numerous questions arose which should be answer to help general quality of the work.

Comment 1: Page 1, line 35-45: The authors cited previously published data of GLOBOCAN from the years of 2018 and 2020. The most recent publication incorporates data of 2022; thus, CA Cancer J Clin. 2024 May-Jun;74(3):229-263 should be referred also.

Response 1: The authors have taken note of the information shared by the reviewer and do wish to indicate that the publication DOI: https://doi-org.ukzn.idm.oclc.org/10.3322/caac.21834 has since been read, referred to and cited in the article.

The authors have also revised a few areas of introduction section to incorporate some of the information garnered from the suggested reference.

See lines 34-41

Globally, there were an estimated 20 million new cancer cases and about 10 million cancer-related deaths in 2022 [1]; these figures remain almost unchanged from the 2020 estimates [2] but demonstrate a notable increase from the 2018 estimates of 18.1 million new cases and 9.6 million cancer-related deaths [3]. In 2022, the combined age-standardized incidence rates for all cancers worldwide were 212.5 and 186.2 for men and women, respectively, while the combined age-standardized mortality rates for men and women were 109.7 and 76.8 per 100,000, respectively [1]. Aside from highlighting the burden that cancer imposes on countries, recent global estimates of incidence and mortality are suggestive of fluctuations in incidence and deaths when compared to 2020 and 2018 estimates [1–3].

Lines 44-47

The GLOBOCAN cancer incidence and mortality reports of 2018, 2020 and 2022 have consistently suggested that female breast, colorectal, prostate and cervical cancers are among the top ten cancers in terms of new cases and cancer deaths in countries worldwide [1–3]. Globally, these four cancers collectively accounted for an estimated 32% of cancer incidence and 23% cancer deaths in 2020 [2], and a 32% cancer incidence and 24% cancer deaths in 2022 [1].

The authors wish to thank the reviewer for urging this response.

Comment 2: Page 2, line 80: The current study focused on the patient material of three institutes, e.g. Sir Lester Bird Medical Centre (SLBMC), The Cancer Centre Eastern Caribbean (TCCEC), and the Medical 81 Benefits Scheme (MBS). If this is a hospital-based assessment, it should be noted in the title of the manuscript.

Response 2: The authors have taken note of the reviewer’s comment and wish to point out that in respect of the study sites used for this study and in respect of what obtains on island,  the Sir Lester Bird Medical Centre (SLBMC), is Antigua and Barbuda’s lone tertiary hospital, The Cancer Centre Eastern Caribbean (TCCEC), a public/private cancer facility (provided external beam radiation therapy plus other services), and the Medical Benefits Scheme (MBS), a statutory health organization, responsible for financing healthcare through subventions and reimbursements for medical services. These centres combined account for a high proportion of cancer-related care as well as contribute to the largest documented evidence of cancer on Antigua and Barbuda.

Notwithstanding this brief explanation, the authors have edited the title to encompass the several contributions of these centres, and on the basis that data was used collectively. Title now reads:

‘Overall Observed Survival of Female Breast, Cervical, Colorectal, and Prostate Cancers in Antigua and Barbuda, 2017-2021: Retrospective Data from Four Study Sites’

Comment 3:

On the other hand, Results section did not describe the coverage of the study compared to the whole domestic population. Note that according to GLOBOCAN 2022, in Barbados ~362 new cases estimated per 100,000 in a single year, while the recent study refers similar amount of cases for 5-year period in Antigua and Barbuda.

Response 3:

The authors have taken note of the reviewers’ comments and wish to share that in respect to describing coverage, we have edited the subsection on ‘general descriptions’ to include an approximation of overall crude incidence based on the 391 combined cancer cases and on the basis of projected population estimates for the country. In this regard we have since enhanced the subsection ‘general descriptions’ by making the following edit and citing the source of population projections for 2017-2021.

See lines 145-150

Generally, a combined total of 391 cases of the four cancers were diagnosed in the period 2017-2021 and were included in the study (Table 1). This approximated to an overall 5-year crude incidence rate of roughly 82.0 (95% CI: 74.0-90.0) per 100,000 population after taking into account local projected population at risk estimates (average of 95,738 per year) for the study period [27].

For the common characteristics examined, the overall mean age at diagnosis was 61.5 (12.9) years, with a range of 24-94 years (Table 1).

Comment 4:

Thus, to avoid further misunderstanding, the Introduction section should share information about the healthcare system of Antigua and Barbuda.

Response 4:

The authors have taken note of the reviewer’s comment and wish to share that we have since incorporated the information re: the importance of the named study sites to cancer care/cancer evidence in the Introduction section so as to address this point as suggested.

See lines 62-68

Concerning cancer care, the Sir Lester Bird Medical Centre is Antigua and Barbuda’s only tertiary care hospital, where most of the island’s cancer cases are diagnosed, treated, and managed. Other areas where cancer care is offered include The Cancer Centre Eastern Caribbean (TCCEC), a public/private cancer facility that provided external beam radiation therapy and other services up until April 2024, and the Medical Benefits Scheme (MBS), a statutory health organization responsible for pharmaceutical care, in addition to financing healthcare through subventions and reimbursements for medical services, among other things [14,15]. Combined, these centers account for a high proportion of cancer-related care and contribute the most to the documented evidence of cancer cases in the country [16].

Comment 5:

According to page 2, line 85, the authors analyzed only the histologically verified cases. However, usually in case of patients with the most advanced stage diagnostic and therapeutic interventions are not performed. In that case the least favorable outcome cases were excluded from the analysis; therefore, the observed survival may be better than real. Firstly, it should be described as one of the limitations. Secondly, the authors should provide information about number of cases with the targeted ICD-10 codes without histological verification.

Response 5: The authors have taken note of the reviewers comments and so as to avoid any misunderstanding re: how cases are diagnosed vis data limitations have opted to remove the word histologically confirmed consistent with the point raised by the reviewer in respect of patients with most advanced stage being excluded from diagnostic/therapeutic options and as well as on the basis of expert guidance received. This meant we have edited the first line of subsection ‘study population.’

Moreover, we have also included as a limitation the issue of cases being excluded re: least favorable outcome’

See lines 340-348

Further, our population of cases could easily have also been impacted by systematic selection bias, particularly in instances where patients who presented with an already advanced disease stage were declined diagnostic and/or therapeutic interventions, were more likely to be underreported, and may have been recommended for disease management elsewhere [52,53]. This could have had the unintended consequence of their data being excluded from the key data capture areas at our study sites, thus affecting the derivation of both overall and cancer-specific study estimates. A future study that accounts for data capture across other areas of cancer care, such as medical outpatient and hospice care, and/or that considers model adjustments to counter the possibility of selection bias may be helpful for deriving more comprehensive survival estimates [54].

Further concerning the call for ‘information about number of cases with the targeted ICD-10 codes without histological verification’ and given that this study relied solely on retrospective data abstracted from medical charts/patient logs, the authors wish to share that because of the scope of the study protocol as well as data limitations this information is not currently available to the authors. The authors have since addressed the possible impact of this in the study’s limitations (see lines referenced above).

See lines 340-348

Comment 6:

In addition, rate of morphologically verified cases is an important quality assurance parameter for cancer registration.

Response 5: The authors wish to thank the reviewer for pointing this out. While we are in agreement that ‘rate of morphologically verified cases is an important quality assurance parameter for cancer registration’, we do wish to share that while steps are currently being put in place for the country to have a cancer registry, at present, there is no established population-based or hospital-based cancer registry in Antigua and Barbuda. Thus, in this case we had to rely on expert guidance which pointed out that other tools besides morphological verification are available to clinicians to aid in making a cancer diagnosis in the local setting, additionally, in our setting there is often reliance on the opinion of two pathologists for cancer diagnosis based on histology or cytology.

Having then considered this limitation, and in conjunction with expert guidance received, we have opted to use the word diagnosed instead of ‘histologically verified’ under subhead ‘study population’ and where appropriate in the manuscript. Additionally, we have since made the necessary edits in our study limitations.

Comment 6:

Page 3, line 96: defining of the applied ICD-10 codes seemed incomplete. Not only C53.9 is related to cervical cancer, but the whole C53 group. C19 (malignant neoplasm of rectosigmoid junction) is absent from the colorectal cancer group, only C18 and C20 was mentioned.

Response 6: The authors wish to thank the reviewer for pointing this out. We have since replaced C53.9 with C53.

Regarding the apparent missing C19, the authors wish to share that the intention of using C18/C20 was our shortened way of saying C18 to C20. To avoid any misunderstanding we have since made the necessary correction. In this regard, C18/C20 is now replaced by C18, C19 and C20.

See lines 91-92

C53-cervical cancer, C18, C19, and C20 -colorectal cancer,

Comment 7:

Based on the Methods section, it was not clear that how persons were identified correctly, how duplications were avoided (e.g. patients were examined or treated in different departments), and how mortality database was linked to hospital data. Is there any unique person ID in the country?

Response 7:

The authors have taken note of the reviewer’s comments and have reinserted a brief explanation that was in the initial draft of the manuscript.

This speaks to the identification of cases across study sites.

See lines

In the absence of a national identification system, data abstracted from study sites were cross-referenced to allow for completeness and the removal of duplicate records where appropriate, by using a mix of their Medical Benefits Scheme identification number (MBS number) for records seen at the Medical Benefits Scheme, and their hospital generated medical patient identifier (MPI) and MBS number for cases at the SLBMC. For cases at TCCEC, we relied on both their TCCEC unique number and MBS number, and for the data on deaths, their MBS number and/or MPI. 

Comment 8:

What was the date of last follow-up when vital status was recorded?

Response 8:

The author have taken note of this observation and have since strengthened the subsection ‘outcomes ascertain’ by including as the last date when vital status was recorded to be the last date of the study period. That is the 31/December/2021.

See line 116

“which was December 31, 2021”

Comment 9:

The authors provided 5-year survival, however, in case of patients diagnosed in 2020 and 2021 it was not complete. Censored cases of those years may impair further survival rate in 2025 and 2026 in sum.

Response 9:

The authors have studied this particular comment of the reviewer very carefully and have taken note of the point posited. Having said that the authors wish to share that it is expressly for the same reason that as part of the analysis we have opted to use the Kaplan-Meier method of analysis.

Our basis for use of this method is premised on literature. For example, according to Goel et al., 2010,

“The Kaplan-Meier estimate is the simplest way of computing the survival over time in spite of all these difficulties associated with subjects or situations.”

Ref:         Goel MK, Khanna P, Kishore J. Understanding survival analysis: Kaplan-Meier estimate. Int J Ayurveda Res 2010;1:274–8. https://doi.org/10.4103/0974-7788.76794.

The authors wish to share that this point observation on the use of Kaplan-Meier analysis vis-à-vis our study was stated as a study strength in our manuscript.

See lines 335-337

Additionally, the use of Kaplan–Meier methodology means that, despite its reliance on univariable analysis, the problem of the short study period was mitigated given its advantage of calculating survival probabilities for cancer at a given point in time within any defined period and/or context of censoring [50].

In addition, the authors have since shared in the study conclusion that the consideration for a much longer study period and more copious data would be beneficial in respect of future cancer survival studies.

See lines  361-362

“and to consider a longer study period and more data as a means of supporting”

Comment 10:

 Page 6: According to Table 2, the observed 5-year overall survival of breast cancer was 44.9%, which is extremely low compared to that of countries with very high human development index. In the literature 70-90% was found, while according to SEER, survival of breast cancer patients with earlies stage exceeds 99% (https://www.cancer.org/cancer/types/breast-cancer/understanding-a-breast-cancer-diagnosis/breast-cancer-survival-rates.html). What is the rationale between these huge differences? Also, the survival of cervical and colorectal cancer showed unfavorable outcome in international context, while prostate cancer was in similar range or mildly exceeded the survival of other countries.

Response 10:

The authors have taken note of the reviewers’ comments and do wish to thank the reviewer for urging a comment from the authors in respect of the points posited.

The authors in responding wish to share that re: ‘survival of breast cancer patients with earlies stage exceeds 99% (https://www.cancer.org/cancer/types/breast-cancer/understanding-a-breast-cancer-diagnosis/breast-cancer-survival-rates.html)’ this material while informative addresses the issue of 5-year relative survival of breast cancer. Our study, cognizant of data limitations, instead addressed the matter of overall observed survival, which according to Mariotto et al., 2014, could be defined as

Overall survival—also called all-cause, observed, and crude survival—is the most easily understood survival measure. It estimates the chance of remaining alive some time after diagnosis.” 

Ref:

Mariotto AB, Noone A-M, Howlader N, Cho H, Keel GE, Garshell J, et al. Cancer survival: an overview of measures, uses, and interpretation. J Natl Cancer Inst Monogr 2014;2014:145–86. https://doi.org/10.1093/jncimonographs/lgu024.

Additionally, and notwithstanding Antigua and Barbuda’s very high human development index, during the period 2017 to 2021, the country’s public health system was void of an active or systematic screening program for the four cancers studied. It was thus on this basis and given the quality of data abstracted that we considered our chosen approach to data analysis.

So as not to lose this important point raised by the reviewer, we have included a paragraph that speaks to HDI vis the absence of national screening.

See lines 321-326

In general, and despite Antigua and Barbuda’s very high human development index [47,48], during the period 2017 to 2021, the country’s public health system notably lacked an active or systematic screening program for the four cancers studied [48]. This means that any observed differences between our study estimates and those generated in the international context, for instance, as in the case of female breast and cervical cancers, could have been severely affected by issues such as (i) the absence or ineffectiveness of early screening and detection initiatives, (ii) access to timely care, and (iii) the level or quality of clinical care and management [49].

Further and concerning the comments “What is the rationale between these huge differences? Also, the survival of cervical and colorectal cancer showed unfavorable outcome in international context, while prostate cancer was in similar range or mildly exceeded the survival of other countries” the authors wish to also state that in the absence of a policy and screening protocol for the named cancers during the study period, this meant that observed differences between our estimates and those of other countries could have invariably been the effects of (i) an absence of early detection and screening, (ii) access to timely care especially for the most vulnerable and under privileged, (iii) the quality of clinical care and management, among other factors.

Similar points have been highlighted by

International Agency for Research on Cancer. International Cancer Survival Benchmarking; SURVMARK: Cancer Survival in High-Income Countries. World Heal Organ 2025. https://survival.iarc.who.int/survmark/about/ (accessed January 21, 2025).

The authors also wish to share that it was only in September 2022, that Antigua and Barbuda instituted an active screening program for cervical cancer. Up until the time of drafting the manuscript national screening programs for the other three cancers mentioned in our study are still lacking.

Comment 11:

The authors should cite international examples for population-based cancer survival studies in the Discussion.

Response 11:    

The authors have taken note of the reviewer’s suggestion and have since included citations re: international examples from, population-based cancer survival studies in the discussion section.

We have since included the below listed references

References

#31

McPherson CP, Swenson KK, Jolitz G, Murray CL. Survival of women ages 40-49 years with breast carcinoma according to method of detection. Cancer 1997;79:1923–32. https://doi.org/10.1002/(sici)1097-0142(19970515)79:10<1923::aid-cncr13>3.0.co;2-x.

#36

Joachim C, Macni J, Drame M, Pomier A, Escarmant P, Veronique-Baudin J, et al. Overall survival of colorectal cancer by stage at diagnosis: Data from the Martinique Cancer Registry. Medicine (Baltimore) 2019;98:e16941. https://doi.org/10.1097/MD.0000000000016941.

#37

Joachim C, Véronique-Baudin J, Razanakaivo M, Macni J, Pomier A, Dorival M-J, et al. Trends in colorectal cancer in the Caribbean: A population-based study in Martinique, 1982-2011. Rev Epidemiol Sante Publique 2017;65:181–8. https://doi.org/10.1016/j.respe.2016.11.002.

#38

Ortiz-Ortiz KJ, Ríos-Motta R, Marín-Centeno H, Cruz-Correa MR, Ortiz AP. Emergency Presentation and Short-Term Survival Among Patients With Colorectal Cancer Enrolled in the Government Health Plan of Puerto Rico. Heal Serv Res Manag Epidemiol 2016;3:2333392816646670. https://doi.org/10.1177/2333392816646670.

Comment 12:

The recent cancer screening system in Antigua and Barbuda may explain some findings of the study.

Response 12:      

The authors also wish to share that it was only in September 2022, that Antigua and Barbuda instituted an active screening program for cervical cancer. A national screening system for the other cancers is still lacking. Notwithstanding this however, some nongovernmental organizations (NGOs) such as Breast Friends and service clubs such as LIONS Club play a role in offering organized breast and prostate cancer screenings, respectively, at least once a year. Data generated by these NGOs are not deposited in any established data repository nor are they available for public research.

This point on the lack of national screening is noted in our discussions section of our manuscript.

The authors do wish to thank the reviewer for the comments offered.

Reviewer 2 Report

Comments and Suggestions for Authors

Major Revisions are needed:

Abstract:

·       Lines 13-15: background is not persuasive!

·       Line 15: Please another word instead of “ascertain!”

·       Line 19: “Survival risk” I nor correct here!

·       Add Number of included patients and their baseline characteristics at the beginning of the result section.

·       The result section needs to be revised to be more clear and also compelling!

Introduction:

·       As GLOBOCAN 2022 is published, update you Ref. 1 and also whole introduction, especially the first paragraph!

(https://acsjournals.onlinelibrary.wiley.com/doi/abs/10.3322/caac.21834)

·       A you examined prognostic factors of survival, it is better to mentioned them in Introduction and discuss it a bit.

Method:

·       Lines 76-87: there is lots of duplicated sentences! Revise it please.

·       Describe how you check the normality of your data and how you interpret it in Statistical analysis, please.

Results:

·       The quality of Figures is too low. Please use higher quality or better formats!

·       Mean of follow-up months should be mentioned!

Discussion:

·       Variables that you analyzed more in result (e.g. parish) should be discussed more here!

In overall:

·       Ensure that all sections adhere to the journal's formatting guidelines.

·       USE simple and common words, instead of complex and unclear ones. Be sure to have a native speaker to revise your work.

Comments on the Quality of English Language

·       USE simple and common words, instead of complex and unclear ones. Be sure to have a native speaker to revise your work.

Author Response

Response to Reviewer 2 Comments

Thank you very much for taking the time to review this manuscript. We do express our appreciations to you for your comments and suggestions offered. It is our hope that the revised manuscript has addressed your concerns. We do look forward to hearing from you on this.

Please find the detailed responses below and the corresponding revisions/corrections highlighted in track changes in the re-submitted files.

2. Point-by-point response to Comments and Suggestions for Authors

REVIEWER 2

General comments:

Major Revisions are needed:

Comment 1:

Abstract:

Lines 13-15: background is not persuasive!

Response 1:    

The authors have taken a careful note of the reviewer’s comment and wish to share that we have since edited the background of the ‘Abstract’ to read as follows:

See lines 13-15

Understanding cancer survival is important for countries like Antigua and Barbuda, where female breast, cervical, colorectal and prostate cancers are burdensome to the healthcare system. This study, therefore, aimed to estimate the survival probabilities of persons diagnosed with these cancers between 2017-2021.

Comment 2:

Abstract:

Line 15: Please another word instead of “ascertain!”

Response 2:    

The authors have taken a careful note of the reviewer’s comment and wish to share that we have since replaced the word ‘ascertain’ with the word ‘estimate.’ This is highlighted in bold below.

Lines 13-15

Understanding cancer survival is important for countries like Antigua and Barbuda, where female breast, cervical, colorectal and prostate cancers are burdensome to the healthcare system. This study, therefore, aimed to estimate the survival probabilities of persons diagnosed with these cancers between 2017-2021.

Comment 3:

Abstract:

Line 19: “Survival risk” I nor correct here!

Response 3:   

The authors have taken a note of the reviewer’s comment and wish to thank the reviewer for pointing this out to us. We have since replaced the words ‘Survival risk’ with the words ‘Hazards ratio.’

See lines 18

‘Hazards ratio’

Comment 4:

Abstract:

Add Number of included patients and their baseline characteristics at the beginning of the result section.

Response 4:  

The authors have taken a careful note of the reviewer’s comment and in thanking the reviewer, do wish to share that we have since included the number of patients and a brief note on their baseline characteristics at the beginning of the result section.

See lines 19-26

A total of 391 diagnosed cases were included in the study (2017-2021): female breast 42%, cervical 10%, colorectal 20%, prostate 28%; Overall, mean age 61.5 (±12.9) years; 62% females; 73% > 55years; 56% from St. John’s, 82% alive at the end of 2021. Median overall survival (years), 4.8-female breast, 4.1-cervical, 4.5-colorectal, unattained (prostate cancer). Cancer-specific overall observed 5-year survival probabilities were 44.9% (female breast), 10.8% (cervical), 19.6% (colorectal), and 69.0% (prostate). Significant association of disease stage with overall survival were observed in female breast and colorectal cancers.

Comment 5:

Abstract:

The result section needs to be revised to be more clear and also compelling!

Response 5:  

The authors have taken a careful note of the reviewer’s comment and in thanking the reviewer, do wish to share that we have since revised the result section to take account of the reviewers suggestion for it to be clear and compelling.

See lines 19-26

A total of 391 diagnosed cases were included in the study (2017-2021): female breast 42%, cervical 10%, colorectal 20%, prostate 28%; Overall, mean age 61.5 (±12.9) years; 62% females; 73% > 55years; 56% from St. John’s, 82% alive at the end of 2021. Median overall survival (years), 4.8-female breast, 4.1-cervical, 4.5-colorectal, unattained (prostate cancer). Cancer-specific overall observed 5-year survival probabilities were 44.9% (female breast), 10.8% (cervical), 19.6% (colorectal), and 69.0% (prostate). Significant association of disease stage with overall survival were observed in female breast and colorectal cancers.

Comment 6:

Introduction:

As GLOBOCAN 2022 is published, update you Ref. 1 and also whole introduction, especially the first paragraph!

(https://acsjournals.onlinelibrary.wiley.com/doi/abs/10.3322/caac.21834)

Response 6:

The authors have taken note of the information shared by the reviewer and do wish to indicate that the publication DOI: https://doi-org.ukzn.idm.oclc.org/10.3322/caac.21834 has since been read, referred to and cited in the article.

The authors have also revised a few areas of introduction section to incorporate some of the information garnered from the suggested reference.

See lines 34-41

Globally, there were an estimated 20 million new cancer cases and about 10 million cancer-related deaths in 2022 [1]; these figures remain almost unchanged from the 2020 estimates [2] but demonstrate a notable increase from the 2018 estimates of 18.1 million new cases and 9.6 million cancer-related deaths [3]. In 2022, the combined age-standardized incidence rates for all cancers worldwide were 212.5 and 186.2 for men and women, respectively, while the combined age-standardized mortality rates for men and women were 109.7 and 76.8 per 100,000, respectively [1]. Aside from highlighting the burden that cancer imposes on countries, recent global estimates of incidence and mortality are suggestive of fluctuations in incidence and deaths when compared to 2020 and 2018 estimates [1–3].

Lines 44-47

The GLOBOCAN cancer incidence and mortality reports of 2018, 2020 and 2022 have consistently suggested that female breast, colorectal, prostate and cervical cancers are among the top ten cancers in terms of new cases and cancer deaths in countries worldwide [1–3]. Globally, these four cancers collectively accounted for an estimated 32% of cancer incidence and 23% cancer deaths in 2020 [2], and a 32% cancer incidence and 24% cancer deaths in 2022 [1].

The authors wish to thank the reviewer for urging this response.

Comment 7:

Introduction:

A you examined prognostic factors of survival; it is better to mentioned them in Introduction and discuss it a bit.

Response 7:  

The authors have taken note of the suggestion of the reviewer and have since inserted a paragraph that briefly speaks to the aspect of prognostic factors.

See lines 52-56

Moreover, these reasons, together with a detailed assessment of prognostic factors, such as patient age and general state of health at presentation, health-seeking behavior, tumor type and size, tumor extent, disease stage, lymph node status, evidence of metastases, evidence of comorbidity, postoperative complications, and the availability of important therapeutic options, generally lend to an enhanced quality of care for cancer patients overall, in addition to predicting the likelihood of disease recurrence and/or death [7–10].

Comment 8:

Method:

Lines 76-87: there is lots of duplicated sentences! Revise it please.

Response 8:  The authors have taken a careful note of the reviewer’s comment and would like to thank the reviewer for pointing out this issue. We do wish to share that we have since revised subsections 2.1 and 2.2 of the Methods section. These subsections now read:

See lines 77-86

2.1 Study design and population

This was a retrospective analytical study comprised of cases of persons, aged 18 years and older, diagnosed with a primary tumour of the female breast, cervix, colorectum, and prostate between January 1, 2017, and December 31, 2021. Cases with recurrent cancer, and a lone case of male breast cancer, were excluded from the study population as their inclusion would have been in contradiction to the study aim.

2.2 Study setting

Study data were obtained by record abstraction using patient files held at the Oncology, Pathology and Urology departments of the Sir Lester Bird Medical Centre (SLBMC), Antigua and Barbuda, The Cancer Centre Eastern Caribbean (TCCEC), and the Medical Benefits Scheme (MBS), Antigua and Barbuda. Data on cancer deaths were obtained from the Ministry of Health, Health Information Division, Antigua and Barbuda.

 Comment 9:

Method:

Describe how you check the normality of your data and how you interpret it in Statistical analysis, please

Response 9:   The authors have taken a careful note of the reviewer’s comment, understanding that the validity of the study results depends on the assessment of normality of the data. We have since included a sentence in the ‘data analysis’ subsection of the Methods section as well as made an edit of the ‘general descriptions’ subsection to account for how normality was assessed.

See lines 128-133

Further, using the continuous variable age, which was common to each cancer-specific dataset and enabled the ease of examination, the distribution of each cancer-specific data was assessed visually for normality using a histogram, normality was also examined with use of the Shapiro-Wilk test for normality [25]. The appearance of a roughly symmetrical or bell-shaped distribution was taken as good judgement that cancer-specific data was normally distributed [25]. Additionally, a Shapiro-Wilk W test p value > 0.05 provided confirmation that each cancer specific data was normally distributed [25].

And see lines 144-145

 “and on the assumption that the data on each cancer type was normally distributed (see supplementary files 1 & 2).”  

We have also included supplementary files (1& 2) to further depict our assessment of normality.

Supplementary file 1: depicts the visual presentation of a normally distributed data by cancer type

Supplementary file 2: depicts the results of the Shapiro-Wilk W test for normality by cancer type

Same as:

Cancer Type

Observations

W

V

Z

Prob >Z

Female Breast

163

0.99

0.61

-1.14

0.873

Cervical

40

0.94

2.18

1.64

0.051

Colorectal

79

0.98

1.46

0.83

0.203

Prostate

109

0.99

0.68

-0.87

0.806

Comment 10:

Results:

The quality of Figures is too low. Please use higher quality or better formats!

Response 10:   The authors have taken a careful note of the reviewer’s comment and have since revised the figures to allow for higher quality images.

See minimized figures below:

Higher quality/better formats of the figures are also given in lines 176-189

Comment 11:

Results:

Mean of follow-up months should be mentioned!

Response 11:  The authors have taken a careful note of the reviewer’s comment and have since made mention of this in the results section and under each cancer-specific subsection.

That was determined to be as follows:

Female breast cancer: 2.41 years (95% CI 2.20-2.63) years          (lines 165)

Cervical cancer: 2.22 years (95% CI 1.77-2.66) years        (lines 204)

Colorectal cancer: 2.25 years (95% CI 1.91-2.59) years   (lines 219)

Prostate cancer: 2.14 (95% CI 1.90-2.39) years      (lines 234)

The authors wish to thank the reviewer for urging this response.

Comment 12:

Discussion:

Variables that you analyzed more in result (e.g. parish) should be discussed more here!

Response 12:  The authors have taken a careful note of the reviewer’s comment and wish to share that we would have already given attention to briefly discussing the variables analyzed in the result section in the discussion section of the manuscript.

See lines: 295-296

, an observation that could be the effect of a probable link between aggressive disease processes and a population composed of 85% of persons of African ancestry [39,40]. 

See lines 297-298

a finding that may suggest that this may be due to persons generally having an almost equal access to cancer care across the country [41].

Comment 13

In overall:

Ensure that all sections adhere to the journal's formatting guidelines.

Response 13:  The authors wish to thank the reviewer for reminding us of this requirement. In our revision/submission we have made the appropriate changes where required.

Comment 14:

In overall:

USE simple and common words, instead of complex and unclear ones. Be sure to have a native speaker to revise your work.

Response 14: The authors wish to thank the reviewer for pointing out this matter. And wish to indicate that in general, we have attempted to replace some words, where in our minds they may cause the average reader to become overwhelmed.

In addition, and so as to improve the quality of the language we referred the paper to MDPI editing services for editing. See certificate in attachment.

The authors do wish to thank the reviewer for the comments offered.

Round 2

Reviewer 1 Report

Comments and Suggestions for Authors

General quality of the manuscript has been improved.

Author Response

ROUND 2

Response to Reviewer 1 Comments

Thank you very much for taking the time to review this manuscript. We do wish to express our thanks to you for the comments your provided in round 1 of the review process which urged us to improve the overall quality of the manuscript considerably.

We did not see any comments for Round 2 of the review process.

The authors do thank you once again for your time and efforts spent conducting our review.

Reviewer 2 Report

Comments and Suggestions for Authors

1- Both in abstract and result: “A total of 391 diagnosed cases were included in the study (2017-2021): female breast 42%, cervical 10%, colorectal 20%, prostate 28%; Overall, mean age 61.5 (±12.9) years; 62% females; 73% > 55years; 56% from St. John’s, 82% alive at the end of 2021. Median overall survival (years), 4.8-female breast, 4.1-cervical, 4.5-colorectal, unattained (prostate cancer). Cancer-specific overall observed 5-year survival probabilities were 44.9% (female breast), 10.8% (cervical), 19.6% (colorectal), and 69.0% (prostate). Significant association of disease stage with overall survival were observed in female breast and colorectal cancers. This is not grammatically correct and should be written in appropriate wording. In this way, it is more similar to a table caption than to the text of the article.

2- This manuscript must be reviewed by a high-quality writer/reviewer, especially for grammar, and writing principles.

3- The quality of the figures are still too low. USE the ORIGINAL ones, not the resized or formatted form of the figures.

Comments on the Quality of English Language

This manuscript must be reviewed by a high-quality writer/reviewer, especially for grammar, and writing principles.

Author Response

ROUND 2

Response to Reviewer 2 Comments

Thank you very much for taking the time to review this manuscript. We do express our appreciations to you for your comments and suggestions offered. It is our hope that the revised manuscript has addressed your concerns. We do look forward to hearing from you on this.

Please find the detailed responses below and the corresponding revisions/corrections highlighted in track changes in the re-submitted files.

2. Point-by-point response to Comments and Suggestions for Authors.

REVIEWER 2

Comment 1:

Both in abstract and result: “A total of 391 diagnosed cases were included in the study (2017-2021): female breast 42%, cervical 10%, colorectal 20%, prostate 28%; Overall, mean age 61.5 (±12.9) years; 62% females; 73% > 55years; 56% from St. John’s, 82% alive at the end of 2021. Median overall survival (years), 4.8-female breast, 4.1-cervical, 4.5-colorectal, unattained (prostate cancer). Cancer-specific overall observed 5-year survival probabilities were 44.9% (female breast), 10.8% (cervical), 19.6% (colorectal), and 69.0% (prostate). Significant association of disease stage with overall survival were observed in female breast and colorectal cancers.” This is not grammatically correct and should be written in appropriate wording. In this way, it is more similar to a table caption than to the text of the article.

Response 1:    

The authors have taken a careful note of the reviewer’s comment, wish to share that having had the manuscript undergone extensive English language editing and after a careful review of the journals guideline re: word limit for abstract, we have since adopted the suggested changes recommended by the MDPI -Language editing services (see copy of certificate 89760 below).

In this regard our abstract now reads:

See lines 14-30

Abstract: Understanding cancer survival is important for countries such as Antigua and Barbuda, where female breast, cervical, colorectal, and prostate cancers are burdensome to the healthcare system. This study therefore aimed to estimate the survival probabilities of patients diagnosed with these cancers between 2017 and 2021. A retrospective analytical study design was used to evaluate cancer cases abstracted from medical records at key study sites. Estimates of observed survival probabilities were determined using a Kaplan–Meier analysis. Significant differences between survival curves were assessed using the log-rank test. Hazard ratios were calculated using Cox regression. P-value <0.05 indicated significance. A total of 391 diagnosed cases were included in this study (2017-2021): female breast cancer accounted for 42%, cervical cancer accounted for 10%, colorectal cancer accounted for 20%, and prostate cancer accounted for 28%. Overall, the mean age of the participants was 61.5 (±12.9) years; 62% were female, 73% were aged > 55 years, 56% were from St. John’s, and 82% were alive at the end of 2021. The median overall survival (years) was 4.8 for female breast cancer, 4.1 for cervical cancer, 4.5 for colorectal cancer, and not reached for prostate cancer. The cancer-specific overall observed 5-year survival probabilities were 44.9% for female breast cancer, 10.8% for cervical cancer, 19.6% for colorectal cancer, and 69.0% for prostate cancer. Significant associations between disease stage and overall survival were observed in female breast and colorectal cancers. This study provides important evidence for the 5-year observed survival probabilities of the studied cancers. Healthcare improvements that support cancer survival are required.

Response 2:    

Considering that comment 1 also speaks to the issue which affected the abstract, also affecting the results, the authors have also taken a careful note of this. We wish to share that on account of feedback/request from the Section Managing Editor of the journal, and in light of the reviewers comments, we have had to incorporate the below change to the subsection ‘general descriptions’ and following extensive language editing using the MDPI -Language editing services (reference certificate 89760)

This section now reads: Lines 181-196

Table 1 presents the overall and cancer-specific baseline details of the study population and on the assumption that the data on each cancer type was normally distributed (Supplementary files 1 and 2). Generally, the sum of 391 cases of the four cancers were diagnosed in the period 2017-2021 and were included in the study (Table 1). This approximated to an overall combined 5-year crude point prevalence rate of roughly 66.8 (95% CI: 59.7-74.6) per 100,000 population after taking into account local projected population at risk estimates (average of 95,738 per year) for the study period [30].

For the common characteristics looked at, the overall median age at diagnosis/presentation was 62 years (Table 1) [20]. Of the defined age categories, the < 55 years age group accounted for 27.4% of cases, with the ≥ 55 years age group accounting for 72.6% of cases (Table 1). By percentage of cases, female breast cancer was responsible  42%, prostate cancer 28%, cervical cancer 10%, and colorectal cancer 20% (Table 1) [20]. Regarding year of presentation, aside from female breast cancer in 2018, the year 2020 had the highest count of diagnosed cancer specific cases (Table 1) [20]. In respect of parishes, St. John’s Parish was responsible for 56% of our case count (Table 1) [20].

Comment 2:

This manuscript must be reviewed by a high-quality writer/reviewer, especially for grammar, and writing principles.

Response 2:   

The authors have taken a note of the reviewer’s comment and wish to thank the reviewer for pointing this out to us. The authors however wish to share that our revised manuscript (after making major revisions based on feedback and/comments from reviewers and Subsection managing editor) did undergo English language editing as per certificate 89760.

Comment 3:

The quality of the figures are still too low. USE the ORIGINAL ones, not the resized or formatted form of the figures.

Response 4:  

The authors have taken a careful note of the reviewer’s comment and in thanking the reviewer, do wish to share that we have since reverted to the original figures. We however had to make an edit to the title of  figure 3 following on from a request for revision received from the managing editor. In the title as well as in the manuscript the words ‘area of residence’ was changed to district. We also wish to share that as per our last submission, our figures as submitted in the attached figures file are of minimum resolution 300 dpi.

We do hope that our submitted figures do meet your acceptance.

Further we have included higher resolution images (300 dpi) in a Zip folder attached with our submission.

See our improved figures below (Lines 224-241 in the manuscript):

See image in attached pdf file

Figure 1. Overall cumulative observed survival by cancer type (a) female breast, (b) cervical, (c) colorectal, and (d) prostate.

 See image in attached pdf file

Figure 2. Overall cumulative observed survival by age-group (a) female breast, (b) cervical, (c) colorectal, and (d) prostate.

See image in attached pdf file

Figure 3. Overall cumulative observed survival by parish (district) (a) female breast, (b) cervical, (c) colorectal, and (d) prostate.

See image in attached pdf file

Figure 4. Overall cumulative observed survival by disease stage (a) female breast, (b) cervical, (c) colorectal, and (d) prostate.

Comment 2:

Comments on the Quality of English Language

This manuscript must be reviewed by a high-quality writer/reviewer, especially for grammar, and writing principles.

Response 2:   

The authors have reviewed the above comments and wish to share that the manuscript was submitted to MDPI language editing services. In our estimation and based on the certificate received, we can assure the reviewer that the service was provided to our satisfaction.

See certificate attached/above (in pdf file).

The authors do wish to thank the reviewer for the comments offered, as they have helped in improving the quality of the manuscript.

Thank you.
